TECHNICAL RELEASE

# A practical DNA data storage using an expanded alphabet introducing 5-methylcytosine

Deruilin Liu[1,2,†], Demin Xu[2,3,†], Liuxin Shi[2,†], Jiayuan Zhang[4], Kewei Bi[5], Bei Luo[6], Chen Liu[6], Yuxiang Li[7], Guangyi Fan[2,8], Wen Wang[2,3,*] and Zhi Ping[2,9,*]

1 College of Life Sciences, University of Chinese Academy of Sciences, Beijing, 100049, China
2 BGI Research, Shenzhen, 518083, China
3 BGI Research, Beijing, 100101, China
4 BGI Research, Hangzhou, 310030, China
5 BGI Research, Changzhou, 213299, China
6 Wuhan BGI Technology Service Co., Ltd., Wuhan, 430073, China
7 HIM-BGI Omics Center, Hangzhou Institute of Medicine (HIM), Chinese Academy of Sciences, Hangzhou, 310030, China
8 BGI Research, Qingdao, 266426, China
9 School of Medicine, The Chinese University of Hong Kong, Shenzhen, Shenzhen, 518172, China

## ABSTRACT

The DNA molecule is a promising next-generation data storage medium. Recently, it has been theoretically proposed that non-natural or modified bases can serve as extra molecular letters to increase the information density. However, this strategy is challenging due to the difficulty in synthesizing non-natural DNA sequences and their complex structure. Here, we described a practical DNA data storage transcoding scheme named R+ based on an expanded molecular alphabet that introduces 5-methylcytosine (5mC). We demonstrated its experimental validation by encoding one representative file into several 1.3~1.6 kbps *in vitro* DNA fragments for nanopore sequencing. Our results show an average data recovery rate of 98.97% and 86.91% with and without reference, respectively. Our work validates the practicability of 5mC in DNA storage systems, with a potentially wide range of applications.

**Availability and implementation:** R+ is implemented in Python and the code is available under a MIT license at https://github.com/Incpink-Liu/DNA-storage-R_plus.

**Subjects** Computer Sciences, Synthetic Biology, Molecular Genetics

Submitted: 02 November 2024

* Corresponding authors. E-mail: wangwen4@genomics.cn; pingzhi@genomics.cn

† Contributed equally.

Preprint submitted at https://doi.org/10.1101/2024.12.26.630439

## INTRODUCTION

In nature, DNA serves as a biological macromolecule comprised of adenine (A), cytosine (C), guanine (G) and thymine (T) ligated by chemical bonds. It has been humanity's primary data storage system for millennia. Currently, the DNA molecule is considered to have extraordinary potential as a novel digital information memory due to its extremely high storage density (estimated physical density of 455 EB/g) [1, 2], remarkable stability (half-life greater than 500 years) [3, 4] and cost-efficient energy consumption [5].

The information density of DNA data storage is closely related to the transcoding schemes. At present, proposed high-performance transcoding schemes, such as YYC [6] and Wukong [7], have improved the information density of standard DNA data storage containing A, C, G and T to 1.95–1.98 bits/base, approaching the theoretical limit of

2 bits/base derived from information theory [8]. Recently, it has been theoretically proposed that expanding the molecular alphabet for DNA data storage by introducing non-natural bases [9, 10] or modified bases [11] can exceed the limits of standard DNA data storage and further increase the information density. Based on the general DNA data storage containing four natural bases, Biswas *et al.* [12] introduced non-natural bases (Ds, Px, Im and Na) into the alphabet of DNA data storage for the first time, and completed experimental validation with actual information density greater than 2 bits/base in 2020. However, the feasibility of the strategy introducing non-natural bases into a molecular alphabet is challenging because of the difficulty of synthesis and non-universality of sequencing methods [13–15] of non-natural DNA sequences. Despite recent progress, uncertainties regarding both the maximum content and the most extended length of homopolymers [16, 17] formed by non-natural bases in DNA sequences add complexity to the DNA data storage system and further restrict its practical applications. From the perspective of compatibility with natural bases, modified bases could be a better option as additional molecular letters of DNA data storage; however, the practicability of DNA data storage based on an expanded molecular alphabet introducing modified bases has not yet been validated.

Here, we report a practical DNA data storage containing 5mC (5-methylcytosine) (DDS-5mC) and develop a direct transcoding scheme named R+ based on an expanded molecular alphabet (EMA). To evaluate the feasibility of DDS-5mC, we showed its experimental validation by encoding a representative file (.txt) into several 1.3~1.6 kbps *in vitro* DNA fragments. Nanopore sequencing results showed an average data recovery rate of 98.97% and 86.91% with and without reference, respectively. Our work should benefit the research and application of DNA data storage based on an expanded molecular alphabet (DDS-EMA).

## RESULTS

### The general principle and feature of R+

Generally, a standard DNA data storage system includes A, C, G and T only. The majority of preceding algorithms or mappings [6, 7, 18–20] on transcoding between digital data and DNA sequences just encompassed two binary bits and four natural bases. Drawing inspiration from the rotational transcoding strategy proposed by Goldman *et al.* in 2013 [21], we described a more universal transcoding scheme named R+. The features of R+ are scalability and inclusivity, which could quickly provide a direct mapping reference between EMA and N-nary digits for researchers when introducing additional molecular letters in the absence of high-performance transcoding strategies. The core of R+ lies in the N-nary rotational transcoding table, as illustrated in Figure 1A, and the size of EMA varies with different N values, leading to distinct limits of information density (Figure 2 and Table 1). We could obtain a quaternary rotational transcoding table by introducing 5mC into the standard molecular alphabet. Before being encoded into DNA sequences, the digital file is typically represented as a string of bytes (numbers between 0 and 255), which is then transformed into base-4, resulting in a string comprising characters from $\{0, 1, 2, 3\}$. The base-4 string is then encoded into a DNA sequence of letters from $\{A, C, G, T, M\}$ using the quaternary rotational table illustrated in Figure 1A. For each digital character to be encoded, the column labelled with the previous letter used and the row labelled by the current digit are selected and encoded using the letter in the corresponding table cell. Notably, as the first digit lacks a previous letter for reference, we designated A as the virtual letter. Taking



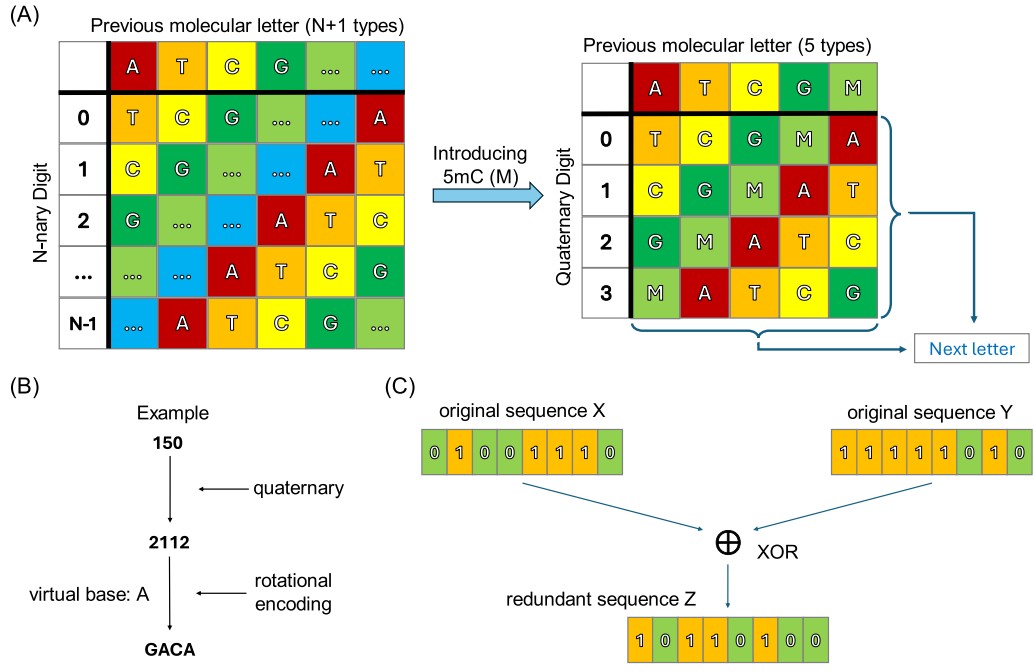

**Figure 1. Principles of R+.**
(A) The rotational transcoding table of R+. Letters A, T, C, and G are adenine, thymine, cytosine, and guanine, respectively, and the letter M represents 5-methylcytosine. (B) The example of converting digital information into DNA sequences using R+. (C) Increasing redundancy by XOR operation.

**Table 1.** The size of the molecular alphabet and the corresponding limit of information density at various $N$ values.

| Value of $N$ | Size of molecular alphabet | Limit of corresponding information density for DNA data storage/bits per base |
|---|---|---|
| 3 | 4 | 2.0 |
| 4 | 5 | 2.32 |
| 5 | 6 | 2.58 |
| 6 | 7 | 2.81 |
| 7 | 8 | 3.0 |
| $N-1$ | $N$ | $\log_2 N$ |

the example shown in Figure 1B, the decimal character 150 is converted to quaternary form first. By referring to the rotational transcoding table, each character in the quaternary string "2112" is mapped to a specific letter, resulting in the final DNA sequence "GACA".

Furthermore, R+ exhibits a dependency on the previous letter to generate the next letter, which leads to a risk of cascading errors caused by potential mutations (substitutions, insertions and deletions) during the decoding process. Therefore, we adopted exclusive-or (XOR) [22] to ensure fidelity. As shown in Figure 1C, every pair of original sequences X and Y can generate a redundant sequence Z through the XOR operation, and any of the three sequences can be recovered by using the other two sequences.

## *In silico* analysis of R+ for stored data recovery

R+ can transcode any format of digital files into DNA sequences in the encoding process and vice versa in the decoding process. To demonstrate the utility of R+, we assessed its

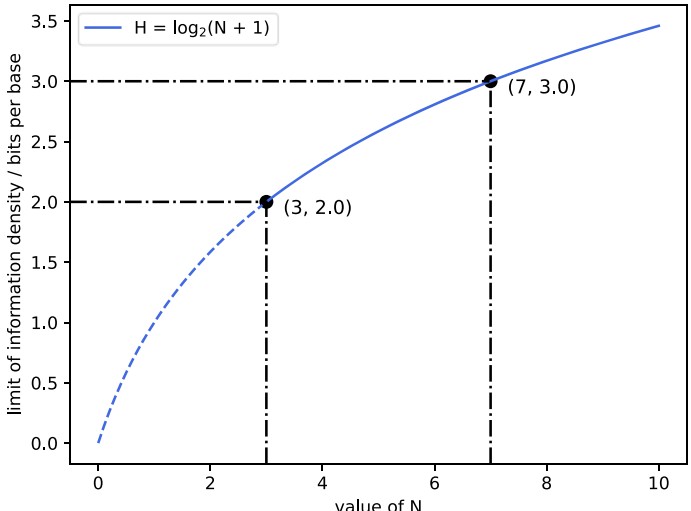

**Figure 2.  The relationship between the value of N and the limit of corresponding information density for DNA data storage.**
From the information entropy formula, we could deduce that the information density limit of DNA data storage (H) has a logarithmic relationship with the size of the molecular alphabet ($N + 1$). When the value of $N$ is 3, the size of the molecular alphabet is 4 (i.e., A, C, G, and T), and the information density of the corresponding standard DNA data storage is limited to 2 bits/base. Similarly, the limit of information density is 3 bits/base when $N$ is equal to 7.

performance in encoding various types of data consisting of documents (text and PDF), programs, images, audio files, and video files. Different data with sizes ranging from about 5.79 KB to 3.99 MB were encoded by R+. The distribution of GC content and homopolymers of DNA sequences encoded from six digital files is shown in Figure 3A, B. As expected, the GC content of the generated sequences fluctuates between 50% and 70%, owing to the fact that letters G, C, and M collectively constitute three-fifths of EMA. Especially, M is a modified form of C resulting in the presence of homopolymers in the generated sequences, which differs from the approach conducted by Goldman *et al.*

Taking into account the influence of mutations on data retrieval in practical validation, we randomly introduced errors into the DNA sequences with different error rates and analyzed the corresponding data recovery rate. The results show that R+ can achieve complete recovery of the digital files by using XOR error correction under an error rate of 0.01% (Figure 4). As illustrated in Figure 3C, the average data recovery rate gradually diminishes with increasing error rates, which could be alleviated through alternative approaches, such as increasing sequencing depth in practical DNA data storage. Furthermore, the coefficient of variation of data recovery rate for the six types of digital files remained at a relatively low level under different simulation test scenarios with varying error rates.

## Experimental validation of R+ by practical data storage

As a proof of concept for practical DDS-5mC, we selected one representative file (poem.txt). We encoded it into 405 DNA strings, one-third of which was the data redundancy generated by the XOR operation to improve data recovery fidelity. Each string contained 100 nt. To meet the required sequence length for nanopore sequencing, we added an 8 nt adaptor

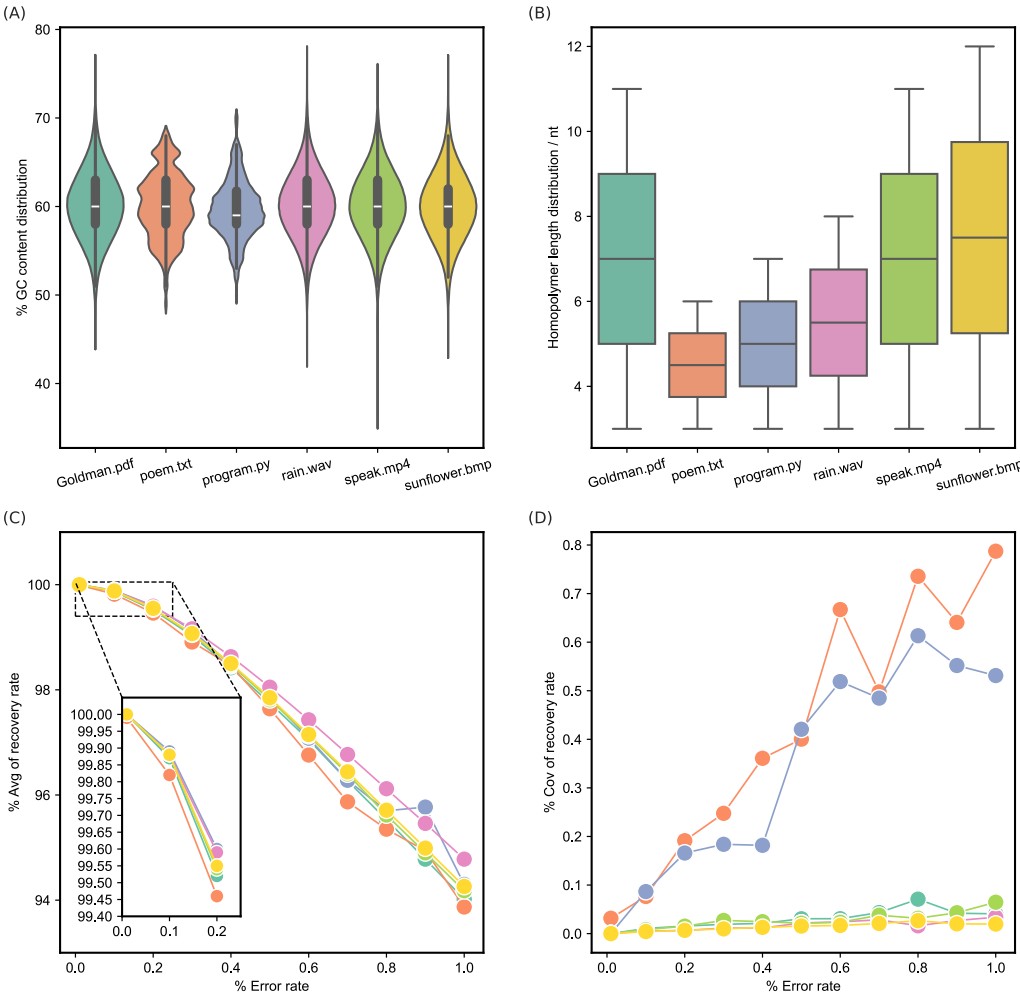

**Figure 3.** *In silico* **analysis for the R+ transcoding scheme.**
Six types of digital files were chosen: a research article (Goldman.pdf, 1.18 MB), four ancient Chinese poems (poem.txt, 5.79 KB), a Python file (program.py, 6.94 KB), an audio clip of rain (rain.wav, 3.81 MB), a video clip of Martin Luther King's speech (speak.mp4, 1.45 MB), and an image of Van Gogh's artwork (sunflower.bmp, 3.99 MB). Among them, digital files at the KB level differ significantly from those at the MB level in aspects such as GC content range and the maximum length of homopolymers (Figure 5). (A, B) The distribution of GC content and homopolymers of DNA sequences encoded by R+ from six types of digital files. (C, D) The average values and coefficient of variations of data bit recovery rates of simulations tests under different error rates. Simulation tests for each error rate were repeated ten times in parallel. Avg: average values; Cov: coefficient of variations.

(Figure 6) to each DNA string for sequential ligation and synthesized oligos corresponding to our designed DNA strings. Next, we assembled a total of 405 synthesized DNA sequences in stages through the ligation, which produced fifteen groups (from A to O in the alphabet) of DNA fragments of 1,620 bp and 1,288 bp, respectively (Table 2). The sequence design and the process of assembly are illustrated in Figure 7A. Ligation products towards each group resulted in a clear band in agarose gel, suggesting that each group could be successfully assembled (Figure 8). Our sequencing and analysis results showed that the acquisition error rate of the original data was 4.95% while the sequencing depth was 10×, and the data recovery rate obtained with and without reference were 98.97% and 86.91%, respectively. Substitution accounted for the largest percentage of data errors in the validation of



**Figure 4. The data recovery rate of six kinds of digital files under different error rates in repeated simulation tests.**
The data recovery rate for the six digital files was almost 100% at an error rate of 0.01%. With the increase in error rate, the data recovery rate in the simulation test decreases gradually.

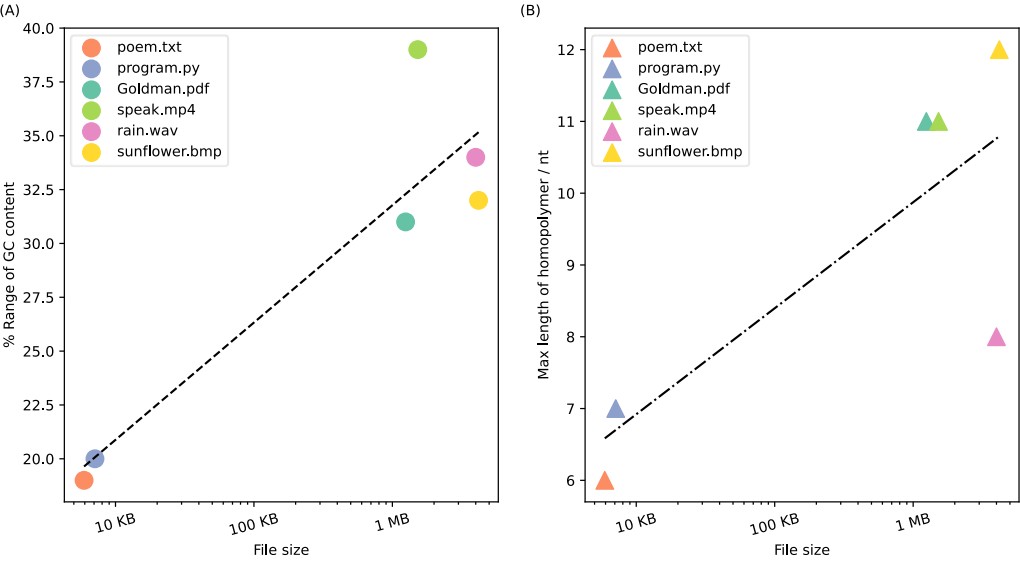

**Figure 5.** **The relationship between the size of the chosen file and attributes of oligos transcoded by R+.** (A) and (B) show the relationship diagrams illustrating the growing tendency of GC content range and the maximum length of homopolymers as the file size increases, respectively.

**Table 2.** The concentration of final products of the step-by-step assembly of DNA sequences.

|                       | A    | B    | C    | D    | E    |
|-----------------------|------|------|------|------|------|
| Oligo 1–15 (ng/μl)    | 6.14 | 28   | 6.02 | 5.52 | 15.5 |
| Oligo 16–27 (ng/μl)   | 4.98 | 8.92 | 4.96 | 9.26 | 44   |
|                       | F    | G    | H    | I    | J    |
| Oligo 1–15 (ng/μl)    | 14.8 | 17.3 | 17.2 | 6.98 | 13.6 |
| Oligo 16–27 (ng/μl)   | 22.8 | 15.8 | 10.4 | 19.1 | 14.3 |
|                       | K    | L    | M    | N    | O    |
| Oligo 1–15 (ng/μl)    | 16.3 | 21.8 | 10.7 | 9.2  | 26.8 |
| Oligo 16–27 (ng/μl)   | 17.5 | 28.8 | 7.4  | 8.9  | 9.2  |

DDS-5mC (Figure 7B), contrasting with the standard nanopore sequencing where indels (insertions and deletions) constitute a relatively significant proportion [23]. It is worth noting that, among all 405 sequences, 377 had substitutions between C and 5mC exclusively (Figure 7C). Considering the dominant usage of the current ONT universal methylation sequencing model in epigenetics research, we speculate this phenomenon primarily arises from the limited accuracy of the universal methylation sequencing model in identifying M at arbitrary positions or non-specific patterns (specific pattern, e.g., cytosine-phosphoric-guanine, CpG), which is inferior to specially designed sequencing model [11]. From the perspective of long-term storage and large-scale applications, the increasing accumulation of substitutions between C and 5mC may compromise the stability and robustness of DNA data storage as the duration and scale expand.

Together, the successful storage of ancient poems in synthetic DNA and their decoding demonstrated the practicability of DDS-5mC, further highlighting the enormous potential of modified bases represented by 5mC in the field of DNA data storage despite the suboptimal recovery rates observed during low-depth sequencing data analysis.

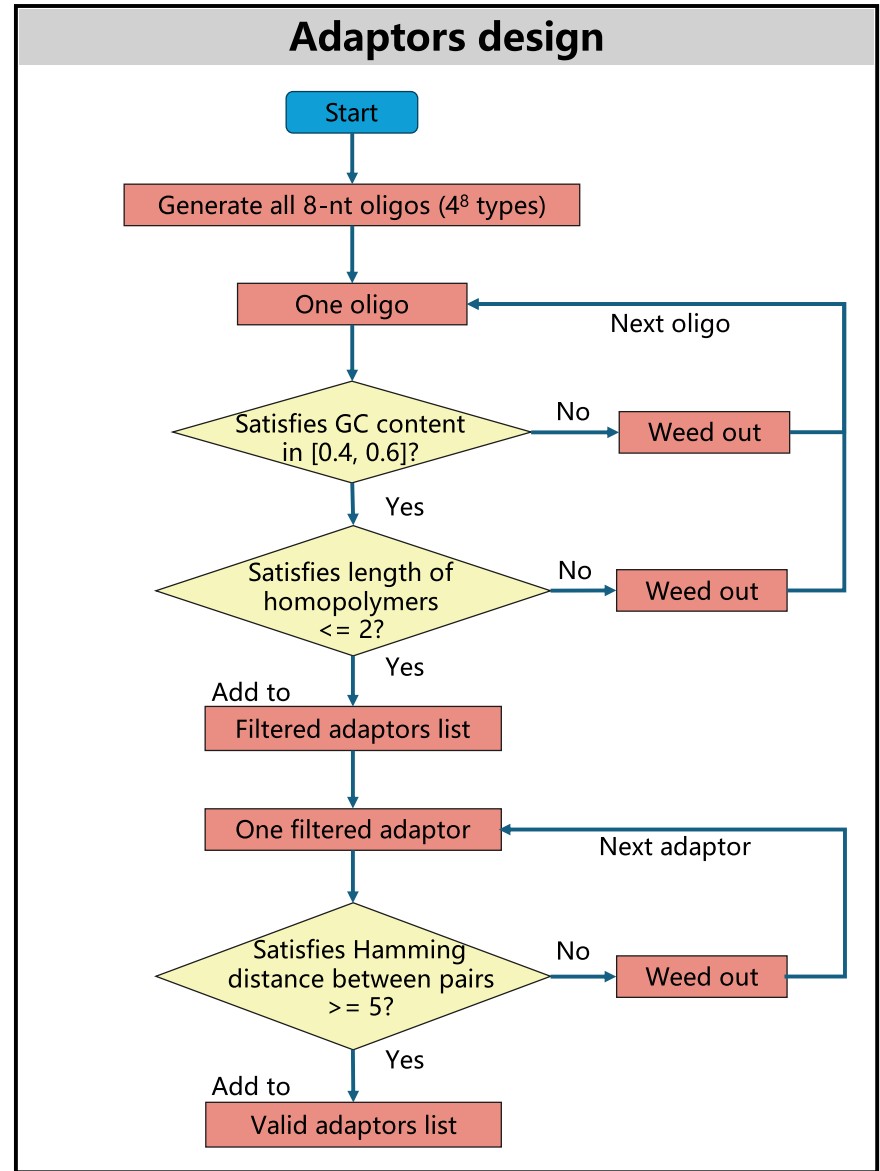

**Figure 6.** **Design flowchart of 8-nt adaptors.**
All 8-nt oligonucleotides as candidate sequences are subjected to a series of filtering criteria (GC content, length of homopolymers, and Hamming distance) to select valid adaptors, avoiding difficulties in biocompatibility and sequence ligation. Possible 8-nt oligos are generated first, totaling $4^8$ types. Next, the oligos undergo a filtering process where only those with a GC content between 40% and 60%, and the maximum homopolymer length not exceeding 2 nt, are retained as candidate adaptors. Finally, pairwise alignments are conducted among the candidate adaptors, and only those with a Hamming distance greater than 4 from all other candidates are valid ultimately.

## DISCUSSION AND CONCLUSION

DDS-5mC introduces 5mC as an extra letter. It increases the limit of information density from 2 ($\log_2 4$) bits/base to 2.32 ($\log_2 5$) bits/base. In general, there are two approaches to raising the information density of DNA data storage without digital data compression. One is to develop more excellent and high-performance transcoding strategies, such as DNA



**Figure 7.** **Experimental validation of data storage using R+ transcoding scheme.**
(A) The sequence design of the oligo generated by R+ and the diagram of oligos' assembly. (B) The proportion of three error types in group A–O. Sub.: substitution; del.: deletion; ins.: insertion. Substitution errors in almost all groups account for more than 80%. (C) The proportion of substitutions between C and 5mC to the total substitutions in each oligos. (D) The percentage error for each oligo position with and without reference. The ratio of substitutions, deletions, and insertions under the condition of no reference is shown in Figure 9.

Fountain [20], YYC [6], and Wukong [7]. Another approach is to introduce new DNA letters into the molecular alphabet. In addition to the non-natural and modified bases mentioned above, composite DNA letters [24] that can substantially reduce the synthesis cycle are also highly promising for enhancing the information density of DNA storage systems. However, the introduction of composite letters involves more complex changes in biochemical

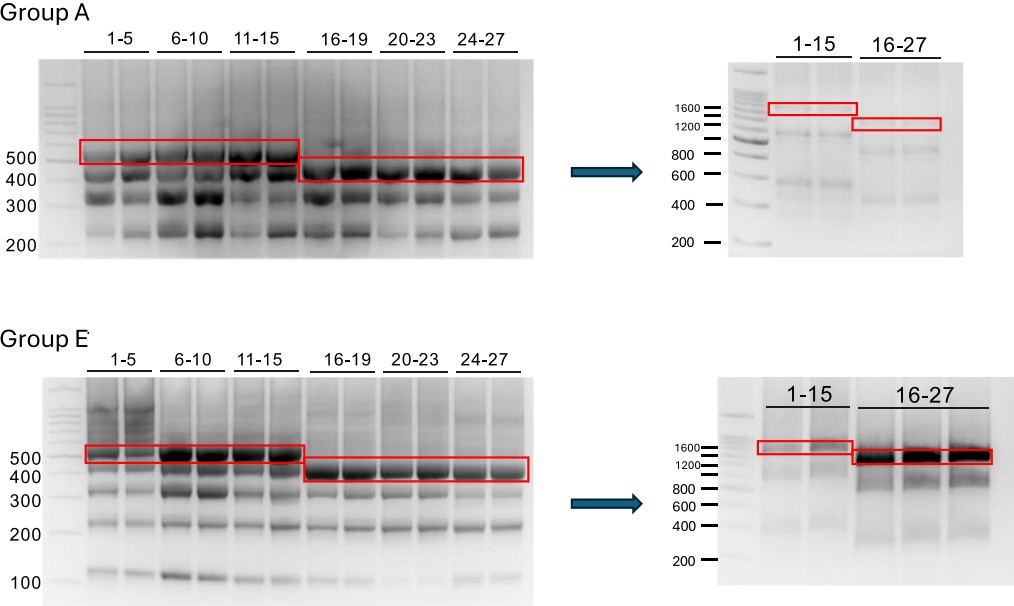

**Figure 8. Gel electrophoresis image of sequence assembly in stages for groups A and E.**
The initial assembly of a group of oligos involves ligating every five of the first 15 oligos and every four of the remaining 12 oligos. The subsequent assembly ligates the first three and the last three products obtained in the previous step.

processes and imposes higher demands on synthesis and sequencing than single letters. Moreover, DDS-5mC employs a developed transcoding scheme R+. The development of R+ is driven by the absence of straightforward methods for DDS-EMA at the moment. Indeed, in addition to the quinary form, R+ can be extended further by adding more modified bases as molecular letters, like N6-methyldeoxyadenosine (6mA) and 7-methylguanine (7mG). Except for the direct mapping relationship between digital data and DNA sequences, this will also provide potential rule generation logic for high-performance transcoding schemes based on EMA, such as "yin" rules in YYC, if required for future research. Nevertheless, because of the simplicity of the generated mapping between digital data and molecular letters, R+ cannot constrain the GC content and homopolymers in encoded DNA sequences, which could pose problems for DNA synthesis. Given the current extensive research on methylation of CpG dinucleotide site in the genome [25, 26] and the sequencing preference of nanopore for methylation at the CpG site, we will subsequently conduct studies on a novel transcoding strategy that is able to yield 5mC-G bindings while keeping a typical GC content (40%–60%) in generated sequences. Moreover, we will select error correction codes with excellent performance, such as the Reed-Solomon (RS) code [18], to enhance the robustness of the novel transcoding scheme.

Despite the demonstration of DDS-5mC, further advancement of DDS-EMA, including DDS-5mC, still encounters certain practical challenges. Firstly, it is difficult to back up long DNA sequences with modified bases for nanopore sequencing by polymerase chain reaction (PCR) amplification. Although long DNA sequences could be backed up by cell passage, additional validation of the host cell is required for long DNA sequences containing modified bases to ensure DNA de-modification (e.g., DNA demethylation [27, 28]) does not occur during the host cell replication cycle. Furthermore, the high synthesis cost of

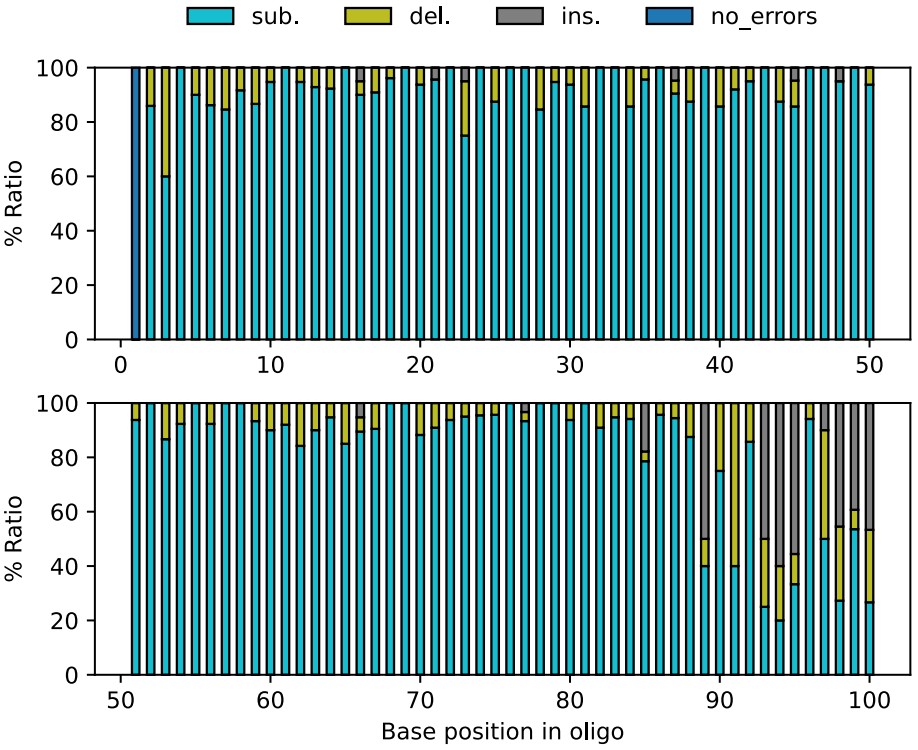

**Figure 9.** The proportion of substitutions, insertions, and deletions for each position in oligo under the condition of no reference.

modified bases represents a significant barrier to its practical application. The current prevailing commercial rates of DNA synthesis result in a cost exceeding $1,500 for storing 1 MB of data in a synthetic oligo-pool. For storing the same volume of data, DDS-EMA with high information density reduces the amount of DNA synthesis. However, given that the synthesis cost of modified bases is approximately two orders of magnitude higher than that of standard bases, this could potentially lead to an overall increase in costs. Recently, a novel storage strategy named "epi-bit" proposed by Zhang *et al.* has partially alleviated the issue related to the synthesis cost of 5mC [29]. They combined molecular movable-type printing with self-assembly-guided enzymatic methylation, thereby decreasing the reliance on conventional synthesis and consequently enhancing both the data writing speed and cost-effectiveness. Moreover, the synthesis cost of modified bases is intrinsically linked to that of natural bases. It is anticipated that as chip-based and enzyme-based DNA synthesis continue to advance, the cost of synthesizing modified bases will experience a significant transformation in the future. Another factor to consider is the accuracy of nanopore sequencing for generated modified bases randomly. As discussed, the existing nanopore-based sequencing model for modified bases exhibits certain pattern biases, likely due to the historical predominance of nanopore sequencing in genomics and epigenetics research. For accurate data reading in DNA storage applications, it is imperative to develop a novel sequencing model tailored to the random methylation patterns generated by the transcoding algorithm. Alternatively, devising a transcoding strategy that accommodates

the inherent pattern biases of the current nanopore sequencing model would also facilitate the practical implementation of modified-base-based DNA storage.

In conclusion, we introduced 5mC as an additional letter into the DNA data storage system based on traditional sequence information and demonstrated the *in vitro* experimental validation of DDS-5mC from data writing to data reading. We also developed a universal transcoding scheme named R+, which is suitable for DDS-5mC. DDS-5mC was mainly constructed to overcome the limitations of non-natural bases as extra molecular letters. The complexity of synthetic pathways, lack of general sequencing methods, and the uncertainty of biological constraints have made non-natural bases unsuitable for practical large-scale applications. On the contrary, the research and synthesis of modified bases (such as 5mC) that can be accurately distinguished by nanopores have reached a relatively mature stage, making them applicable as extra letters in the molecular alphabet for DNA data storage [30–32]. Our experimental validation results of DDS-5mC further indicate the feasibility and practicability of DDS-EMA with modified bases. In recent years, practical applications of DNA data storage in both everyday and specialized scenarios [33–35] have been validated successfully. With the continuous advancements in synthesis and sequencing technologies, DDS-EMA is poised to facilitate the realization of practical DNA data storage across various scenarios as a higher information-density storage strategy.

## METHODS

### *In silico* simulation

All encoding, decoding, and error analysis were performed in a Windows 10 environment running on an i5 central processing unit with 16 GB of random-access memory, using Python 3.10.

The test files included six types of digital files: .pdf, .txt, .py, .bmp, .wav, and .mp4. The segment length of the binary information was set to 22 bytes. All the test files were transcoded to a DNA string using R+'s code and inserting XOR redundancy. Each DNA string was analyzed to obtain the overall distribution of GC content and length of homopolymers. Additional simulation tests were performed to evaluate the influence of mutations on data recovery. The generated DNA strings were artificially introduced with different error rates (from 0.01% to 1%), then decoded into binary information, and aligned with the original digital file to derive data recovery rates at different error rates.

### Assembly of 5mC fragments

The assembly of 5mC fragments was accomplished through a two-step T4 ligation process (see protocols.io protocol (Figure 10) [36]). Before ligation, 108-nt oligonucleotides were phosphorylated using T4 Polynucleotide Kinase (NEB, CAT#: M0201L), followed by annealing to form double-strand DNA with 8-nt sticky ends. Next, every set of five 108-bp fragments was ligated to form a 540-bp segment. Thereafter, the three 540-bp segments were further assembled using T4 DNA ligase (NEB, CAT#: M0202L). The phosphorylation and annealing were performed using the following program: 37 °C for 2 h, followed by 95 °C for 5 min, 70 °C for 30 min, 55 °C for 5 min, 37 °C for 5 min, and final hold at 12 °C. The T4 ligation was performed at 16 °C for 18 h.

For gel electrophoresis, the two-step T4 ligation used 3% and 1.5% agarose gels, respectively, which were run at 180 V for 60 min. The ligation products were purified by gel purifications before usage. The gel purifications were performed using the Freeze N

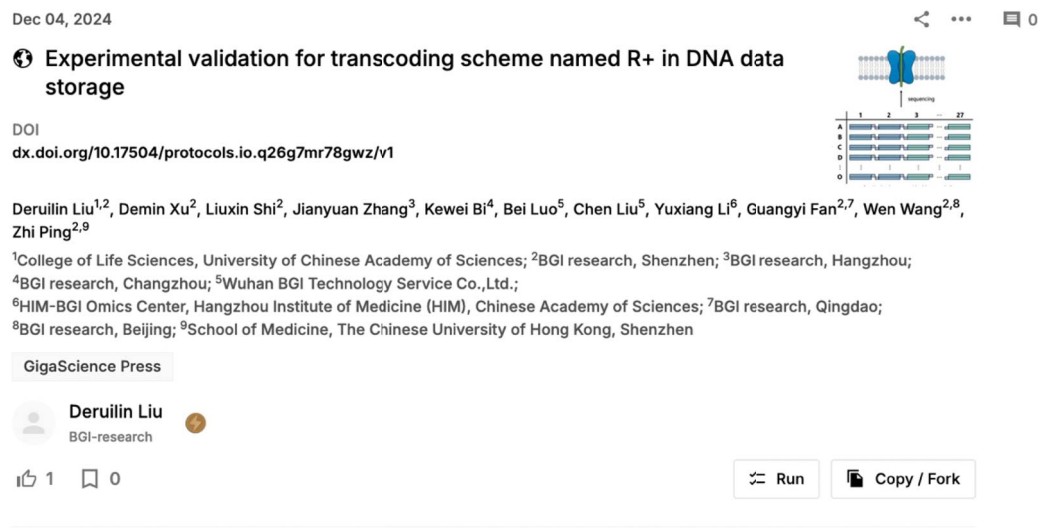

**Figure 10.** Protocols.io protocol for experimental validation of the transcoding scheme named R+ in DNA data storage [36]. https://www.protocols.io/widgets/doi?uri=dx.doi.org/10.17504/protocols.io.q26g7mr78gwz/v1

Squeeze DNA Gel Extraction Spin Columns (BIO-RAD, CAT#: 4106139 and NucleoSpin Gel and PCR Clean-up Kit (MACHEREY-NAGEL, CAT#: 740609.250).

## 5mC fragments sequencing experiments on PromethION

All the sequencing experiments with 5mC fragments were done on the PromethION platform (RRID:SCR_017987) using R10.4.1 flow cells [36]. The library construction components were provided in the SQK-NBD114.96 kit developed by ONT for ligation and sequencing. Ligation was performed using NEB Blunt/TA Ligase Master Mix (catalogue number M0367) and NEBNext Quick Ligation Module (catalogue number E6056); all the NEB reagents needed were included in the Native Barcoding Kit. Before the sequencing adapter ligation, the pooled barcoded sample was prepared with up to 96 unique barcodes for every sample to be combined and run together on the same flow cell. The barcode ligation reaction was stopped by adding EDTA. The final sequencing library mixture contained 12 µl of eluted DNA, 37.5 µl of sequencing buffer, and 25.5 µl of library beads. Loading 75 µl of the library mix to the flow cell via the SpotON sample port after the flow cell was initially flushed with priming mix via the priming port. The raw data were recorded in the FAST5 file format using MinKNOW software (Oxford Nanopore Technologies).

## Statistical analysis of experimental validation

Data were not pre-processed unless explicitly declared. By using a Python script, the methylation tag information was obtained from the .bam file to determine the probability of each cytosine being methylated in the sequence (0–1 probabilities were mapped to integers 0–256, with a threshold of 128 where values above were considered methylated). Sequencing reads containing 5mC were obtained based on this probability. Subsequently, all reads were aligned with the reference for initial screening, and only reads with an identity value greater than or equal to 0.85 were retained. Next, ONT nanopore barcodes at both ends and assembly adapters of each identified read were removed to generate several oligos ranging in length from 80 nt to 120 nt. Multiple sequence alignment was then

performed using 10× oligos, and the consensus sequence from the alignment was extracted as the ultimate DNA sequence. Error statistics were obtained by pairwise alignment between the consensus and the reference, and the final data recovery rate was defined as the ratio of accurately decoded bits to the total bits in the original file.

Methylation data parsing was mostly performed in STOmics DCS Cloud. Some tools and Python extension packages were used for these works, which included SAMtools and pysam. Sequencing data analysis, visualization analysis, and visualization were mostly performed in JetBrains PyCharm Professional 2024.2.3 using Python version 3.10. Some tools and Python extension packages were utilized for the process as well, which included clustalw2, numpy, pandas, openpyxl, matplotlib, seaborn, and biopython.

## AVAILABILITY OF SOURCE CODE AND REQUIREMENTS

- Project name: R plus
- Project home page: https://github.com/Incpink-Liu/DNA-storage-R_plus
- Operating system(s): Windows 10
- Programming language: Python 3.10
- License: MIT License
- RRID: SCR_026005
- Bio.tools ID: R_plus.

The source code for R+ and the drawing script for all figures are available in the GitHub URLs mentioned above. The corresponding package is made available at https://pypi.org/project/R-plus, and the users can install this package by command "pip install R-plus".

## DATA AVAILABILITY

The source data for all figures and snapshots of the code are available in GigaDB [37]. The sequencing raw data that support the findings of this study have been deposited in the EBI (Biostudies ID: S-BSST1792) and CNSA (https://db.cngb.org/cnsa) of the CNGBdb with accession code CNP0006533.

## LIST OF ABBREVIATIONS

5mC, 5-methylcytosine; A, adenine; C, cytosine; DDS-5mC, DNA data storage containing 5mC; DDS-EMA, DNA data storage based on an expanded molecular alphabet; EMA, expanded molecular alphabet; G, guanine; PCR, polymerase chain reaction; T, thymine; XOR, exclusive-or.

## DECLARATIONS

### Ethical approval

Not applicable.

### Competing interests

All of the authors are employees of BGI. The authors otherwise declare that they have no competing interests.

## Authors' contributions

DL, WW and ZP mainly developed the idea and the algorithm and drafted the manuscript and figures; DX mainly did the molecular experiments and prepared the samples; DL, LS, JZ, YL, and GF mainly implemented data analysis and prepared the related figures; KB mainly did the synthesis of oligonucleotides with modified bases; BL and CL mainly did the sequencing of oligonucleotides with modified bases. DL, WW and ZP drafted and revised the manuscript; WW and ZP supervised the work jointly.

## Funding

This work was supported by the National Key Research and Development Program of China of YL (2021YFF1200204), the National Natural Science Foundation of China to ZP (no. 32101182) and WW (no. 32201175), and the Shenzhen Science, Technology and Innovation Commission grant no. SGDX20220530110802015 to ZP.

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
