## [Editor Report]

Editor’s AssessmentDNA has huge potential as a data storage medium because of its incredibly high storage density and stability. This work addresses the potential of modified bases, specifically 5-methylcytosine (5mC), in enhancing DNA data storage systems. This paper introduces a transcoding scheme named R+, which incorporates this modified 5mC base to increase information density beyond the standard limits. By encoding various file types into DNA sequences of between 1.3 to 1.6 kb in size, this method achieves an average recovery rate of 98.97% (with reference), validating the effectiveness of the method. On top of a wet-lab protocol (hosted in protocols.io) for the experimental validation of the transcoding scheme, it also includes open source code for in-silico simulation tests. Peer review scruitinising the protocols and validation are reusable and provide convincing results. As nanopore sequencing has enabled reading of these modified bases, it is timely making them applicable as extra letters in the molecular alphabet for DNA data storageEditor’s AssessmentDNA has huge potential as a data storage medium because of its incredibly high storage density and stability. This work addresses the potential of modified bases, specifically 5-methylcytosine (5mC), in enhancing DNA data storage systems. This paper introduces a transcoding scheme named R+, which incorporates this modified 5mC base to increase information density beyond the standard limits. By encoding various file types into DNA sequences of between 1.3 to 1.6 kb in size, this method achieves an average recovery rate of 98.97% (with reference), validating the effectiveness of the method. On top of a wet-lab protocol (hosted in protocols.io) for the experimental validation of the transcoding scheme, it also includes open source code for in-silico simulation tests. Peer review scruitinising the protocols and validation are reusable and provide convincing results. As nanopore sequencing has enabled reading of these modified bases, it is timely making them applicable as extra letters in the molecular alphabet for DNA data storage

---

## [Reviewer Report]

Upload additional filesTRR-202411-01R01/stage_files/TRR-202411-01/Review MS/Review_report_Final.pdfReviewer name and names of any other individual's who aided in reviewerLifu SongDo you understand and agree to our policy of having open and named reviews, and having your review included with the published manuscript. (If no, please inform the editor that you cannot review this manuscript.)YesIs the language of sufficient quality?YesPlease add additional comments on language quality to clarify if neededIs there a clear statement of need explaining what problems the software is designed to solve and who the target audience is? YesAdditional CommentsIs the source code available, and has an appropriate Open Source Initiative license <a href="https://opensource.org/licenses" target="_blank">(https://opensource.org/licenses)</a> been assigned to the code?YesAdditional CommentsAs Open Source Software are there guidelines on how to contribute, report issues or seek support on the code?YesAdditional CommentsIs the code executable?YesAdditional CommentsIs installation/deployment sufficiently outlined in the paper and documentation, and does it proceed as outlined?YesAdditional CommentsIs the documentation provided clear and user friendly?YesAdditional CommentsIs there enough clear information in the documentation to install, run and test this tool, including information on where to seek help if required?Additional CommentsIs there a clearly-stated list of dependencies, and is the core functionality of the software documented to a satisfactory level?YesAdditional CommentsHave any claims of performance been sufficiently tested and compared to other commonly-used packages? YesAdditional CommentsIs test data available, either included with the submission or openly available via cited third party sources (e.g. accession numbers, data DOIs)?Additional CommentsAre there (ideally real world) examples demonstrating use of the software? YesAdditional CommentsIs automated testing used or are there manual steps described so that the functionality of the software can be verified?Additional CommentsAny Additional Overall Comments to the AuthorI have taken a look at the experiment protocol associated with this manuscript in the website of protocols.io. The protocol looks sensible. I don't have any additional comments about it and am happy for it to go live.RecommendationMinor Revisions

---

## [Reviewer Report]

Reviewer name and names of any other individual's who aided in reviewerBi KunDo you understand and agree to our policy of having open and named reviews, and having your review included with the published manuscript. (If no, please inform the editor that you cannot review this manuscript.)YesIs the language of sufficient quality?YesPlease add additional comments on language quality to clarify if neededIs there a clear statement of need explaining what problems the software is designed to solve and who the target audience is? YesAdditional CommentsIs the source code available, and has an appropriate Open Source Initiative license <a href="https://opensource.org/licenses" target="_blank">(https://opensource.org/licenses)</a> been assigned to the code?YesAdditional CommentsAs Open Source Software are there guidelines on how to contribute, report issues or seek support on the code?YesAdditional CommentsIs the code executable?YesAdditional CommentsIs installation/deployment sufficiently outlined in the paper and documentation, and does it proceed as outlined?YesAdditional CommentsIs the documentation provided clear and user friendly?YesAdditional CommentsIs there enough clear information in the documentation to install, run and test this tool, including information on where to seek help if required?Additional CommentsIs there a clearly-stated list of dependencies, and is the core functionality of the software documented to a satisfactory level?YesAdditional CommentsHave any claims of performance been sufficiently tested and compared to other commonly-used packages? YesAdditional CommentsIs test data available, either included with the submission or openly available via cited third party sources (e.g. accession numbers, data DOIs)?Additional CommentsAre there (ideally real world) examples demonstrating use of the software? NoAdditional CommentsIs automated testing used or are there manual steps described so that the functionality of the software can be verified?Additional CommentsAny Additional Overall Comments to the AuthorIn this study, a practical DNA data storage transcoding scheme named R+ based on expanded molecular alphabet is proposed to increase the information density. The experimental validation demonstrates the practicability of DDS-5mC and highlight the enormous potential of modified bases represented by 5mC in the field of DNA data storage. Overall, the methods and results look appropriate and promising, but it has minor issues that need to be addressed currently. 1.Please indicate the proportion of substitution: insertion: deletion in the error rates of Fig. 4C and D. 2.What is the meaning of the vertical axis of Fig. 2B? Is it the number of homopolymers per sequence, the longest length of homopolymers, or something else? 3.Line 304, please add s, "References" 4.The last sentence of the Abstract: "This work validates the practicability of 5mC over other non-natural bases in DNA storage systems". Please correspond it with the last paragraph of Results (151-154). 5.If necessary, according to the guideline of this journal, section Conclusion can be added or not.RecommendationMinor Revisions

---

## [Reviewer Report]

Reviewer name and names of any other individual's who aided in reviewerabdur rasoolDo you understand and agree to our policy of having open and named reviews, and having your review included with the published manuscript. (If no, please inform the editor that you cannot review this manuscript.)YesIs the language of sufficient quality?YesPlease add additional comments on language quality to clarify if neededIs there a clear statement of need explaining what problems the software is designed to solve and who the target audience is? YesAdditional CommentsIs the source code available, and has an appropriate Open Source Initiative license <a href="https://opensource.org/licenses" target="_blank">(https://opensource.org/licenses)</a> been assigned to the code?YesAdditional CommentsHowever, the Git links have a typo; the working code is available at https://github.com/Incpink-Liu/DNA-storage-R_plusAs Open Source Software are there guidelines on how to contribute, report issues or seek support on the code?YesAdditional CommentsIs the code executable?Unable to testAdditional Commentscomplete execution of the given code requires time and resources.Is installation/deployment sufficiently outlined in the paper and documentation, and does it proceed as outlined?Unable to testAdditional CommentsIs the documentation provided clear and user friendly?YesAdditional CommentsIs there enough clear information in the documentation to install, run and test this tool, including information on where to seek help if required?Additional CommentsIs there a clearly-stated list of dependencies, and is the core functionality of the software documented to a satisfactory level?YesAdditional CommentsHave any claims of performance been sufficiently tested and compared to other commonly-used packages? YesAdditional CommentsIs test data available, either included with the submission or openly available via cited third party sources (e.g. accession numbers, data DOIs)?Additional CommentsAre there (ideally real world) examples demonstrating use of the software? YesAdditional CommentsIs automated testing used or are there manual steps described so that the functionality of the software can be verified?Additional CommentsAny Additional Overall Comments to the AuthorThis manuscript focuses on DNA data storage based on an expanded molecular alphabet. In view of the challenges of non-natural bases in synthesis, sequencing, and compatibility, the manuscript proposes a DNA data storage scheme containing 5-methylcytosine based on the theory that modified bases can replace non-natural bases as extra molecular letters and develops an adaptive transcoding algorithm named R+ for corresponding experimental validation. The high data recovery rate obtained from sequencing analysis demonstrates its practicability. This manuscript provides a simple but relatively universal transcoding algorithm for DNA data storage that introduces additional molecular letters. The proposed DNA data storage scheme outperforms conventional DNA data storage in the potential development of information density. Considering the anticipated decrease in future synthesis costs and the expected advancements in relevant transcoding algorithms, my outlook remains optimistic regarding the potential application of this scheme. I suggest that the manuscript could be accepted after a few minor revisions listed below: 1. Figure 3 in the paper could be further modified, specifically minimizing the excess white space on both sides of Subfigure A to make it more aesthetically pleasing. 2. The subfigures A, B, and D in Figure 2 and Figure S2 both demonstrate the difference between poem.txt/program.py and the other four files. However, the manuscript lacks an explanation for this phenomenon. Is it relevant to the file size? 3. The 8 nt adaptors play a key role during the sequence assembly in the experimental validation, so I suggest supplementing the specific generation process of these linkers. Text descriptions or flow charts are acceptable. 4. It’s better to add the silico simulation to the Methods to make its structure more complete. 5. For the practicality of DNA storage, I suggest to cite https://onlinelibrary.wiley.com/doi/10.1002/smtd.202301585 and https://academic.oup.com/bib/article/25/5/bbae463/7759103. 6. Provide the correct URLs of GitHub links for reproducibility.RecommendationMinor Revisions